# Comparison of SARS-CoV-2 Evolution in Paediatric Primary Airway Epithelial Cell Cultures Compared with Vero-Derived Cell Lines

**DOI:** 10.3390/v14020325

**Published:** 2022-02-05

**Authors:** Connor G. G. Bamford, Lindsay Broadbent, Elihu Aranday-Cortes, Mary McCabe, James McKenna, David G. Courtney, Olivier Touzelet, Ahlam Ali, Grace Roberts, Guillermo Lopez Campos, David Simpson, Conall McCaughey, Derek Fairley, Ken Mills, Ultan F. Power

**Affiliations:** 1Wellcome-Wolfson Institute for Experimental Medicine, Queen’s University Belfast, Belfast BT9 7BL, UK; L.Broadbent@qub.ac.uk (L.B.); mmccabe783@qub.ac.uk (M.M.); david.courtney@qub.ac.uk (D.G.C.); o.touzelet@qub.ac.uk (O.T.); a.ali@qub.ac.uk (A.A.); g.roberts@qub.ac.uk (G.R.); G.LopezCampos@qub.ac.uk (G.L.C.); david.simpson@qub.ac.uk (D.S.); 2Medical Research Council-University of Glasgow Centre for Virus Research, University of Glasgow, Glasgow G61 1QH, UK; elihu.aranday-cortes@glasgow.ac.uk; 3Regional Virus Laboratory, Belfast Health and Social Care Trust, Belfast BT12 6BA, UK; james.mckenna@belfasttrust.hscni.net (J.M.); conall.mccaughey@belfasttrust.hscni.net (C.M.); derek.fairley@belfasttrust.hscni.net (D.F.); 4Patrick G. Johnston Centre for Cancer Research, Queen’s University Belfast, Belfast BT9 7AE, UK; k.mills@qub.ac.uk

**Keywords:** SARS-CoV-2, virus evolution, primary airway epithelial cells, Vero cells

## Abstract

SARS-CoV-2 can efficiently infect both children and adults, albeit with morbidity and mortality positively associated with increasing host age and presence of co-morbidities. SARS-CoV-2 continues to adapt to the human population, resulting in several variants of concern (VOC) with novel properties, such as Alpha and Delta. However, factors driving SARS-CoV-2 fitness and evolution in paediatric cohorts remain poorly explored. Here, we provide evidence that both viral and host factors co-operate to shape SARS-CoV-2 genotypic and phenotypic change in primary airway cell cultures derived from children. Through viral whole-genome sequencing, we explored changes in genetic diversity over time of two pre-VOC clinical isolates of SARS-CoV-2 during passage in paediatric well-differentiated primary nasal epithelial cell (WD-PNEC) cultures and in parallel, in unmodified Vero-derived cell lines. We identified a consistent, rich genetic diversity arising in vitro, variants of which could rapidly rise to near fixation within two passages. Within isolates, SARS-CoV-2 evolution was dependent on host cells, with paediatric WD-PNECs showing a reduced diversity compared to Vero (E6) cells. However, mutations were not shared between strains. Furthermore, comparison of both Vero-grown isolates on WD-PNECs disclosed marked growth attenuation mapping to the loss of the polybasic cleavage site (PBCS) in Spike, while the strain with mutations in Nsp12 (T293I), Spike (P812R) and a truncation of Orf7a remained viable in WD-PNECs. Altogether, our work demonstrates that pre-VOC SARS-CoV-2 efficiently infects paediatric respiratory epithelial cells, and its evolution is restrained compared to Vero (E6) cells, similar to the case of adult cells. We highlight the significant genetic plasticity of SARS-CoV-2 while uncovering an influential role for collaboration between viral and host cell factors in shaping viral evolution and ultimately fitness in human respiratory epithelium.

## 1. Introduction

Severe acute respiratory syndrome coronavirus 2 (SARS-CoV-2) (Family: *Coronaviridae*; Genus: *Betacoronavirus*) has a positive-sense, non-segmented, single-stranded RNA genome of ~30,000 nucleotides in length [1,2]. The SARS-CoV-2 genome encodes at least 29 proteins, expressed from translation of a 5′ major open reading frame (ORF1ab), including Nsp3 and Nsp12 (viral RNA-dependent RNA polymerase), and a series of nested transcripts at the 3′ terminus, including Spike (S; the viral attachment and fusion glycoprotein) and ORF7a. SARS-CoV-2 emerged in the human population in late 2019, causing coronavirus disease 2019 (COVID-19) [3]. Reflecting its likely zoonotic origins, SARS-CoV-2-like and other SARS-related viruses have been detected and isolated from horseshoe bats and pangolins from Asia [4]. Early pandemic SARS-CoV-2 was highly transmissible, with an R0 of up to ~5, and with a moderate case fatality rate (~1%), especially pathogenic in elderly or individuals with co-morbidities [5]. However, significant disease can occur in children and young people, as well as the presence of potentially fatal post-infection syndromes [6]. Although efforts to control the impact of SARS-CoV-2 infection include the use of recently developed safe and effective vaccines [7] and therapeutics such as dexamethasone [8], COVID-19 continues to exert a significant clinical burden across the world.

SARS-CoV-2 productively infects the epithelial cells lining the upper and lower respiratory tract, including those in the nasal cavity and the alveoli of the lung [9]. By virtue of interaction with Spike, SARS-CoV-2 exploits host cell protein angiotensin-converting enzyme 2 (ACE2) as its receptor [10]. Additionally, for entry to occur, Spike requires activation by two host proteases, furin and transmembrane protease serine 2 (TMPRSS2)-like proteases, which cleave Spike at the S1/S2 boundary between its two subunits (S1 and S2) and the S2′ site in S2, allowing the release of the fusion peptide at target cell membranes [11]. Following binding to ACE2, a proteolytically activated Spike can fuse the viral envelope with the host cell membrane, releasing the infectious genome into the cytoplasm.

A seemingly unique feature of SARS-CoV-2 is the presence of a polybasic cleavage site (PBCS) at the S1/S2 boundary that is cleaved by furin, enhancing TMPRSS2-mediated activation [11]. Loss of the polybasic cleavage site (PBCS) has been demonstrated to reduce transmission and virulence of SARS-CoV-2 in animal models [12,13]. Since its initial emergence, SARS-CoV-2 has continued to evolve and adapt to the human population with several putatively beneficial mutations arising in Spike, such as D614G and N501Y, that affect Spike stability and binding to ACE2, and antibody-escape mutations in the amino-terminal domain (NTD) [14]. Together, these mutations of interest are found in constellations in so-called variants of concern (VOC) such as Alpha or Delta, which represent strains of SARS-CoV-2 with evident phenotypic differences, such as enhanced transmissibility, pathogenicity and/or reduced sensitivity to antibody-mediated neutralisation in humans [14].

As SARS-CoV-2 continues to spread, and interventions and vaccines are being rolled out, there remain significant unknowns as to how SARS-CoV-2 may adapt further to humans. Knowledge of the genetic and molecular correlates of this difference in transmissibility is crucial for the understanding of CoV pandemic preparedness and informing strategies for surveillance and control. In vitro models can help disentangle the factors affecting evolution, identify new ones and highlight mutational tolerance. Here, we undertook a side-by-side comparison of SARS-CoV-2 evolution by whole-genome sequencing of two isolates, grown in parallel in standard Vero-derived cells and paediatric human well-differentiated primary nasal epithelial cells (WD-PNECs), which are a useful model for probing virus–host interactions in the respiratory tract [9,15,16]. Our data demonstrate that paediatric WD-PNEC cultures restrain SARS-CoV-2 evolution compared to Vero (E6) cells, disclosing clear roles of both viral and host cell factors in shaping SARS-CoV-2 genetic and functional changes as well as identifying molecular features required for efficient infection of primary cells.

## 2. Materials and Methods

### 2.1. Continuous Cell Line Culture

In this study, 3 continuous cell lines were used: Vero wildtype (number), VeroE6 and VeroE6 expressing human ACE2 and TMPRSS2 (VAT) [17]. Of note, standard VeroE6 cells were only used for the passage of PHE. All cells were grown in DMEM (5% FCS *v*/*v*) with antibiotics. VAT cells were maintained in the presence of additional antibiotics to select cells carrying transgenes. Cell lines were routinely tested for mycoplasma contamination and no evidence of contamination was detected.

### 2.2. WD-PNECs

Nasal epithelial cells from preschool age children with recurrent wheeze (for initial passaging) and from healthy adults (for final comparison of PHE and BT20.1 P4 viruses) were obtained by brushing of the nasal turbinates with an interdental brush (DentoCare, London, UK). Cells were cultured in monolayer until passage 3, then seeded onto collagen-coated Transwells (6 mm, 0.4 µm pore size; Corning). Once confluent, the apical medium was removed to create an air–liquid interface which, together with specialised media (Pneumacult ALI, Stemcell Technologies), triggered differentiation [18,19]. Complete differentiation (after a minimum of 21 d) was confirmed by an intact culture, extensive cilia coverage and mucus production.

### 2.3. Viruses

Two SARS-CoV-2 isolates were used throughout this study: PHE and BT20.1. PHE was provided as an early passage isolate on VeroE6 cells while BT20.1 was provided directly as a nasopharyngeal swab in virus transport media clinical material from a positive case from Belfast in June 2020. Stocks were prepared in Vero or VeroE6 cells in DMEM containing 2.5% FCS (*v*/*v*) infected at a low MOI (~0.001). Infections were harvested when maximal cytopathic effect was noted, usually between 3–4 days post infection. Infected culture supernatant was harvested, clarified by centrifugation and stored at −80 °C. WD-PNECs were apically infected with SARS-CoV-2 for 1 h, after which the inoculum was removed and the apical surface was gently rinsed with DMEM. The virus was harvested from WD-PNECs by incubation of the apical surface with DMEM for 5 min at room temperature in the absence of serum. The harvested virus was immediately stored at −80 °C. All SARS-CoV-2 work was carried out under BSL3 conditions in a dedicated facility in QUB.

### 2.4. Plaque Assays

Our plaque assay protocol is based on the methodology available here: https://www.protocols.io/view/viral-titration-of-sars-cov-2-by-plaque-assay-semi-be4zjgx6 (accessed on 7 July 2020). Near confluent monolayers of Vero cells in 24- or 6-well plates were infected. On the day of titration, the growth media was replaced with DMEM (0% FCS) (250 µL). Virus dilutions were prepared on plates and incubated for 30 min, after which the 2× overlay medium (containing 2% agarose) was added. Plates were incubated for 3 days at 37 °C. At 3 dpi, PFA (8%) was added to the cultures and cells were fixed/inactivated for at least 20 min. Following fixation, the PFA was removed and monolayers were stained for 10 min using crystal violet (1% *w*/*v* in ethanol 20%). Following staining, residual crystal violet solution was removed and plates were rinsed in water and submerged in Chemgene prior to drying and removing from the hood for visualisation and quantification. To calculate PFU/mL, plaques at a dilution were quantified, the precipice of this number used, and multiplied by the dilution factor (4). For visualisation of plaque assays, whole plates were scanned using a Celigo imaging cytometer (Nexcelom Bioscience).

### 2.5. Virus Whole-Genome Sequencing

Virus whole-genome sequencing used methods developed by the ARTIC network (https://artic.network (accessed on 7 December 2021); [20]) and the COG-UK Consortium. Culture supernatants were inactivated by addition of Triton X-100 to 1.5% (*v*/*v*). Viral RNA (total nucleic acid) was extracted from inactivated samples (200 µL) using the MagNA Pure Compact instrument and MagNA Pure Compact Nucleic Acid Isolation Kit I (Roche Molecular Systems Inc, Burgess Hill, UK). Purified nucleic acid was eluted into 100 µL and used immediately or stored at −80 °C. For first-strand cDNA synthesis, nucleic acid (5 µL) was used as the template for reverse transcription using LunaScript^®^ RT SuperMix Kit (New England Biolabs, Hitchin, UK) in 20 µL reaction volume. Primers were annealed (65 °C, 5 min, snap-cool on ice) prior to the addition of reverse transcriptase. Reactions were incubated at 42 °C (50 min) and then stopped at 70 °C (10 min). The resulting cDNA was used immediately for PCR or stored at −20 °C. In brief, these were run as two separate multiplex PCR “pools” (A and B) using the ARTIC version 3 primer set (ARTIC nCoV-2019 V3 Panel, IDT DNA Inc., Leuven, Belgium; https://github.com/artic-network/primer-schemes/tree/master/nCoV-2019, accessed on 7 December 2021) and Q5 DNA polymerase mastermix (New England Biolabs, Herts, UK). Following PCR, the amplicons from pools A and B were combined, and the resulting pooled amplicons (98 × 450 bp overlapping tiled amplicons, spanning the SARS-CoV-2 genome) were purified using Kapa HyperPure beads (Roche Molecular Systems Inc.) and quantified using a Qubit fluorometer and dsDNA HS Assay Kit (Thermo Fisher Inc., Manchester, UK). Amplicon sequencing libraries were prepared using the Nextera DNA Flex library kit according to the manufacturer’s instructions (Illumina Ltd., Cambridge, UK). Libraries were sequenced on a MiSeq (Illumina, Essex, UK) using a MiSeq Reagent Kit v2 and 2 × 151 bp paired-end sequencing protocol (Illumina).

### 2.6. Sequence Analysis

The FASTQ files were uploaded to the Galaxy web platform, and we used the public server at usegalaxy.eu to analyse the data [21]. The workflow used was specially optimised for Illumina-sequenced-based ARTIC pair end data with the intention to detect allelic variants (AVs) in SARS-CoV-2 genomes [22]. This analysis converted FASTQ data into annotated AVs through a series of steps that include QC, trimming ARTIC primer sequences off reads with the iVar package, mapping using bwa-mem, deduplication, AV calling using lofreq, and filtering AVs that both occurred at an allele frequency (AF) ≥5%, and were supported by ≥10 reads. As we could not determine the background frequency of mutations, we focused on the variants with a minor allele frequency ≥5%, and those that were supported by ≥10 reads in at least one passage of the series. Furthermore, we focused our greater analysis on those found in more than one passage and those that substantially rise in frequency. Raw sequencing data are available via online repositories (European Nucleotide Archive) linked: 28 July 2021|PRJEB46668 (ERP130880)|“Comparison of Sars2 evolution in vero derived versus primary human airway cells”.

## 3. Results

### 3.1. Isolation and Passage of SARS-CoV-2 in Unmodified Vero-Derived Cells

To begin to understand the evolution of SARS-CoV-2, we first needed to generate characterised stocks of virus (Figure 1a). In the first instance, a low passage isolate (passage 1, p1) of SARS-CoV-2 (England 02/20) was obtained from Public Health England (PHE) and is referred to as PHE. This stock was from a sample isolated on VeroE6 cells and represents one of the earliest isolates of SARS-CoV-2 in the UK during the pandemic. PHE is from clade A and does not contain the D614G substitution in Spike (Appendix A) [23,24]. Upon receipt, we carried out a further three passages on VeroE6 cells passaging at an MOI of 0.001, harvesting stocks at 96 hpi when extensive cytopathic effects were observed. The PHE strain grew efficiently, reaching titres of >10^6^ pfu/mL (Figure 1b) and was cytopathic, inducing “webbing” and cell rounding, consistent with previous reports (data not shown).

As we wanted to understand the viral factors that may drive evolution, and since results obtained from only one isolate may be non-representative, we next obtained an independent—but comparable—clinical nasal/pharyngeal swab sample containing SARS-CoV-2, which we termed BT20.1 (Belfast 06/20). This strain represents an isolate from the UK’s “first wave” and is a representative of clade B that contains the D614G mutation in Spike (Appendix A). Unlike PHE, BT20.1 was isolated on standard Vero cells (CCL-81) and passaged to P4 (multiplicity of infection (MOI)~0.001 and passaged every 3 days). Like PHE, BT20.1 grew efficiently, reaching titres of >10^6^ pfu/mL (Figure 1b).

Both PHE and BT20.1 formed plaques on standard Vero cells in all passages (Figure 1c,d). Comparison of plaque sizes between P2 and P4 identified differences in plaque size composition following Vero cell passage. This was most evident for PHE, which became predominantly large plaques (Figure 1e,f). As observed in the plaque edge, BT20.1 induced consistent cell-to-cell fusion, unlike PHE (Appendix A).

### 3.2. Sequencing of SARS-CoV-2 Passage Series in Unmodified Vero-Derived Cells

As we had successfully generated comparable in vitro passage series for two relevant isolates of SARS-CoV-2, we next determined what genetic changes, if any, occurred during passage in unmodified Vero-derived cells. Whole-genome sequences of our SARS-CoV-2 stocks at each passage were generated and minor sequence analysis (>5% minor allele frequency) was carried out, comparing variations present in the Wuhan-Hu-1 reference (NC_045512.2) genome sequence for SARS-CoV-2 (Appendix A). Unfortunately, the sequence depth and quality were not sufficient to reconstruct whole-genome sequences for BT20.1 P1 isolate material, likely due to insufficient viral material resulting from the initial isolation. Therefore, we focused our analysis on PHE P1-4 and BT20.1 P2-P4.

Analysing mutations in the PHE passage series, we identified four changes (C8782T; T18488T; T28144C; A29596G) relative to Wuhan-Hu-1 consistently at ~100% at all passages, likely reflecting fixation in the original virus stock (Figure 2a). These changes were considered intrinsic to that particular strain and were not analysed any further herein as we wished to focus on variants arising during passage. Sequencing confirmed the presence of D614 in Spike, consistent with it being an early SARS-CoV-2 isolate.

With the core changes described above, two major mutations were observed: a synonymous (T23605G) and non-synonymous out-of-frame deletion (deletion of 24 nucleotides AATTCTCCTCGGCGGGCACGTAGTG 23597A; resulting in the replacement of nine amino acids (679–687; NSPRRARSV) in Spike with an isoleucine (I)) mapping to the polybasic cleavage site (PBCS) (Figure 2a). Deletion of the PBCS ablated the T23605G synonymous variant in the process. This occurred at P3, although the deletion was observed in the original P1 material from PHE. Furthermore, we detected 15 minor variants (non-consensus) that had an allele frequency (AF) of >5% in at least one sample of the passage series. These changes mapped to several genes and proteins of SARS-CoV-2, including ORF1AB, Spike, E, N and ORF10 (Appendix A). Interestingly, we observed a cluster of three mutations occurring in the amino terminal domain (NTD) of Spike, appearing at P3 and rising in frequency at P4. Two of these Spike NTD mutations were similar to mutations occurring in VOCs: D215G and an out-of-frame deletion of 24 nucleotides (GCTATACATGTCTCTGGGACCAATGGTA21761G), resulting in a loss of nine amino acids IHVSGTNGT (aa67-76). Additionally, we noticed a convergent mutation of L37 in E, detecting two mutations resulting in L37F and L37R. To determine the reproducibility of passage sequencing, an independent P4 PHE (P4B) was generated from P3 and sequenced, with very high levels of similarity between the two (Appendix A).

Like PHE, we identified core changes inherent to BT20.1 (Figure 2b), which were greater in number than PHE (ten vs. four), consistent with its later isolation (February 2020 vs. June 2020) (Appendix A). These changes included, but were not limited to, D614G in Spike; R203K and G204K in N; and an out-of-frame deletion of five nucleotides in ORF7A, leading to its premature truncation. Like PHE, we identified mutations arising rapidly upon consecutive passage in Vero cells (i.e., were not detected at P2), including the non-synonymous mutations T293I in NSP12 and P812R in Spike. Both mutations had similar patterns of change in frequency and constituted the majority of sequences by P3. Like PHE, we also detected minor variants (nine), including G1251V and S1252C in Spike.

### 3.3. Passage of SARS-CoV-2 in Paediatric Primary Human Airway Epithelial Cell Cultures

We next sought to investigate the effect of host cell type on subsequent viral evolution, as our previous analysis assessed the contribution of viral background to viral evolution in unmodified Vero-derived cells. To this end, in parallel, we passaged SARS-CoV-2 samples on well-differentiated primary human airway epithelial cell cultures until P4, in a similar protocol as was carried out in Vero cells (Figure 3a). Primary cultures included WD-PNECs derived from two paediatric donors. For both PHE and BT20.1, robust infection and passage on WD-PNECs were established. For PHE, WD-PNECs were initially infected at MOI of 0.1 and the virus was harvested at 2–3 dpi, using the original P1 virus material. This was repeated for BT20.1, except unlike PHE, BT20.1 was directly isolated on primary cultures from the obtained clinical material. SARS-CoV-2 grew well in the primary cultures, reaching titres of ~10^6^ pfu/mL in 2–3 days in the apical compartment. Samples at each passage were subjected to sequencing as outlined above and analysed in a similar manner to those from the Vero cell passage series. For BT20.1, only P2, P3 and P4 were sequenced to compare with the data available for the equivalent Vero passage series.

In contrast to what was observed in VeroE6 cells, we did not detect any major genetic changes in PHE following passage in WD-PNECs (Figure 3b). However, we did identify the PBCS deletion at low levels in minor variant analysis, but never reaching majority. Together with PBCS, we found 34 changes as minor variants. From passage to passage, these mutations appeared and disappeared stochastically. Similarly, in BT20.1, unlike the Vero cell passage, we did not find corresponding mutations in Nsp12 or Spike (Figure 3c). However, we identified a single amino acid deletion, Y1595, in NSP3. Intriguingly, this variant was maintained throughout the passage series at a moderate frequency of ~45%. SARS-CoV-2 was titrated by plaque assay on Vero cells during the passage series (Appendix A). We noticed slightly reduced titres of BT20.1 in primary cells at P4 compared to earlier passages, which was not observed for PHE (Appendix A). WD-PNEC-grown viruses had less obvious plaques (Appendix A) and no evidence of cell-to-cell fusion was identified, even for BT20.1, which was fusogenic when grown in Vero cells (Appendix A). Similar to passage in Vero cells, we identified two mutations in Spike (G1251V and S1252C), which appeared at low frequencies (<10%) and never increased (Appendix A).

### 3.4. Phenotypic Differences between SARS-CoV-2 PHE and BT20.1 P4

Our data showing host cell dependency in viral evolution suggest differential fitness for specific viral genotypes (e.g., Vero cell-derived mutations that were not observed in WD-PNECs were less fit in primary cells). To test this hypothesis, we focused subsequent analysis on PHE and BT20.1 Vero P4 stocks with clear genetic differences between them, including the PHE PBCS deletion in Spike, and the P812R (Spike) and NSP12 mutations in BT20.1. To this end, we wished to directly compare the growth and multi-cycle replication kinetics of both strains in cell culture models of infection. To achieve this, we carried out a comparison of growth kinetics in several cell culture models, including Vero cells, VeroE6 cells modified to express human ACE2 and TMPRSS2 (VAT) [17] and WD-PNECs (adult nasal) (Figure 4a–c). Of note, unmodified Vero and VeroE6 cells do not express human ACE2 and have very low levels of TMPRSS2 [25]. In Vero cells, SARS-CoV-2 grew to peak extracellular infectivity titres by ~48 hpi with titres of ~10^6^ pfu/mL. We noticed a growth attenuation of BT20.1 in Vero cells compared to PHE (Figure 4a). Comparing virus growth in VAT cells (Figure 4b), both viruses grew better, but the relative attenuation of BT20.1 was not observed in VAT cells. In contrast to previous Vero cell experiments, we observed a prominent growth defect (100–1000-fold differences) of PHE compared to BT20.1 at early time points during infection (24/48 h) in WD-PNECs. However, both viruses reached similar titres by 72 hpi (Figure 4c), which could suggest that the PBCS is not essential for WD-PNEC infection, but more work is needed to address this. Together these data clearly demonstrate phenotypic differences between our Vero cell-passaged viruses, indicating a critical role for the PBCS for efficient replication in primary cells.

## 4. Discussion

Investigation of the patterns of SARS-CoV-2 genetic diversity worldwide during outbreaks has already facilitated a genetic-based nomenclature of lineages and has also highlighted the emergence of functionally relevant mutations, such as D614G in Spike and those contained in extant VOCs [14]. Complementary to this, in vitro systems are an incredibly useful and tractable means to understand the forces influencing this viral evolution, particularly those that mimic in vivo-relevant conditions, such as WD-PNECs [9]. However, much of our understanding using in vitro models of SARS-CoV-2 evolution is from cell cultures derived from adults. Therefore, questions remain concerning the consequences of passage in paediatric cultures, especially as children have distinct outcomes following SARS-CoV-2 infection [27,28,29,30,31].

Our data from paediatric WD-PNEC cultures and Vero cells, as well as that of others, demonstrate significant standing genetic diversity in viral populations in vitro that can be acted upon by rapid evolutionary processes. Although also observed in our work, from early in the study of SARS-CoV-2 evolution in Vero-like cells, it was revealed that the virus readily diversifies during culture, with the most evident being mutations mapping to the PBCS of Spike [23,32,33,34,35]. Consistent with our work, other studies have identified enhanced genetic stability, particularly of the PBCS only, during passage of one strain in Calu3 or adult primary airway organoids [33]. One notable finding of our work, which builds upon previous studies, is that on several occasions for both isolates, we observed a rapid increase in frequency of specific mutations in the PBCS and independently of it (PBCS deletion and P812R in Spike, and T293I in NSP12) to near fixation over the course of a couple of passages in Vero cells. These patterns suggest a selective phenotypic advantage in the Vero cell culture system. Similar changes (including P812R and the NTD deletion) were identified in other studies [36,37]. The fact that identical mutations arise independently (e.g., loss of the PBCS and P812R) is highly suggestive of convergent evolution, perhaps toward a similar phenotype. Our work on the loss of the PBCS in Vero cells and its association with attenuation in WD-PNECs is consistent with previous reports. However, it is noteworthy that we did not observe deletion or mutations in or around the PBCS during passage of BT20.1 in Vero cells.

In addition to the loss of the PBCS, we observed P812R in Spike and T293I in NSP12, although we were not able to associate them with changes in virus growth in WD-PNECs due to a lack of an additional comparable wild-type isolate. However, the fact that parallel passage in WD-PNECs did not result in their increased frequency suggests that they confer a subtle hitherto unrecognised disadvantage in the primary epithelial cell system. P812R is a non-conservative change and rapidly rose to near fixation alongside NSP12 in BT20.1 in Vero cells. P812 sits near the S2′ cleavage site and is a highly conserved position in SARS-CoV-2. However, non-P residues (e.g., serine) are occasionally found in nature but are rare (https://nextstrain.org/, accessed on 7 December 2021), which suggests a functional defect in vivo. Interestingly, P812R was observed before, in at least two other studies, associated with a change in Spike activity using infectious SARS-CoV-2 and one using a chimeric vesicular stomatitis virus encoding SARS-CoV-2 Spike [36,37]. Like previous work, we noted an association of P812R with enhanced cell-to-cell fusion when BT20.1 grown on Vero cells is compared to that grown in WD-PNECs (Appendix A). It was suggested that P812R generated a novel PBCS at the S2′ site [37]. Cleavage by furin-like proteases could thus compensate for the lack of TMPRSS2-mediated proteolysis and activation in Vero cells. Although it is possible that P812R confers a similar phenotypic change as the PBCS deletion, it is not likely to be identical, given the clear differences in growth between PHE and BT20.1 in Vero cells and WD-PNECs. Along with P812R in S, BT20.1 carried a mutation in NSP12 (T4685I/T293I), which is the viral RNA-dependent RNA polymerase. The mutation sits on the surface in close proximity to a zinc-binding site of the interface domain that mediates intra-NSP12 interactions and interactions between NSP12 and other polymerase co-factors, such as NSP8 [38]. Given the linkage between P812R and T4685I, further molecular virological work using isogenic viruses generated through reverse genetics is required to ascertain the impact of this mutation in relevant cell models. The mutation T4685I arose with P812R, possibly suggesting genetic linkage, although this remains to be determined.

It is of interest that BT20.1 carries a deletion in ORF7A that results in a frameshift and C-terminal truncation of the protein, likely ablating the transmembrane domain and tail. ORF7A is a type 1 transmembrane protein, and it has numerous putative functions involved in host–pathogen interactions and immune evasion [39]. ORF7A truncations in SARS-CoV-2 isolates have been discovered before, possibly associated with reduced capacity to subvert the innate immune response [39]. However, the previous work was carried out using non-clinically relevant cell models, such as Vero or HEK-derived lines. Our work suggests that full-length ORF7A is not required for replication in Vero or WD-PNECs and likely serves an accessory function that may affect replication and/or transmission in particular circumstances.

Not only did we observe changes reaching near fixation in our dataset, we also identified several lower-frequency mutations in our viral populations. Consistent with this variation within a stock, we also noticed plaque size variation in passage stocks, suggestive of functional differences between viral sub-clones (Figure 1e,f). We detected an in-frame deletion of a single codon in the C-terminus of NSP3, located in the Y1 domain, which is located on the cytoplasmic face of the virus-remodelled ER membrane, where it may regulate replication complex stability by interacting with NSP4 [40]. NSP3 itself is a multifunctional protein involved in numerous viral processes. The fact that the deletion did not rise to fixation suggests that it is at a competitive disadvantage compared to wild-type. The mutation in NSP3 is also interesting because it is maintained at a moderate frequency. Of considerable interest is the overlap between variations observed in Vero cells and that of VOCs, especially in the NTD of Spike. We observed three mutations in the NTD in PHE P3 and P4: E180K, D215G and a deletion resulting in the loss of nine amino acids. Variants identified in this study mapping to the ectodomain of Spike are marked on a structural model (Appendix A). For D215G and the deletion, these mutations are similar to those in VOCs, such as Alpha and Beta variants. Regarding mutations in NTD loops, several VOCs have convergently modified the amino acid identity of the loop. While in vivo, this may be the result of antibody selection, in our system, there are no antibodies, which suggests a role for NTD mutations independent of antibody selection. The rise in frequency is suggestive of a fitness advantage of these mutations. Further work is required to determine the function of the NTD of Spike and the impact of these mutations on the virus life cycle.

While general trends were similar between our two isolates in Vero cells (i.e., mutations rising to high frequency), specific mutations observed were not. It is likely that the evolution of key mutations reflects inherent biological differences in viruses and not subtle changes in passaging conditions. In PHE, loss of the PBCS occurred, which was not observed in BT20.1, and vice versa regarding P812R and NSP12. This is consistent with an effect dependent on viral input or strain or genetic background through epistatic interactions between mutations, such as D614G in Spike. However, in numerous reports, isolation and passage of SARS-CoV-2 on Vero cells resulted in a loss of the PBCS, which was not observed in our BT20.1 passage series. Alternatively, the mutation P812R could functionally achieve the same phenotype as the deletion of the PBCS, although our primary cell infection model, in which PHE was attenuated compared to BT20.1, would suggest that this is not the case. Finally, one additional interpretation is based on chance effects, which could be better delineated through the use of more isolates grown in Vero cells.

By comparing the evolution of the same isolates in two distinct cell culture systems, we observed a dependence on host cell substrate on downstream virus evolution. Namely, passage in WD-PNECs resulted in enhanced stability of SARS-CoV-2 genetic diversity at the consensus level. While the PBCS, P812R and NSP12 changes were identified in PHE and BT20.1 when grown in Vero cells, these changes did not rise to high frequencies in WD-PNECs. Differential accumulation of mutations may reflect distinct host cellular environments encountered upon passage in Vero or WD-PNECs. This reflects major differences in these cell substrates, including (i) species and tissue differences, (ii) reduced levels of TMPRSS2 in Vero cells and (iii) reduced innate immune response in Vero cells, as they are deficient in type 1 interferon production [41]. Relevant to this is the role of PBCS/TMPRSS2 interaction in evasion of SARS-CoV-2 restriction factor interferon-induced transmembrane 2 (IFITM2) [42]. However, what affects the rise in the P812R/NSP12 mutation remains unknown. Future work will assess the effect of these changes in BT20.1 upon replication in WD-PNECs. Additionally, during passage of BT20.1 in WD-PNECs, we identified a deletion in Nsp3, although the relevance and mechanism of this change are unknown. Reference [43] is cited in the Appendix A.

In conclusion, by studying the evolution of SARS-CoV-2 during passage in distinct cellular substrates (including paediatric airway cultures), we shed light on the forces that shape viral fitness, unveiling a collaboration between both viral and host factors in driving SARS-CoV-2 genetic diversity, which helps define the molecular correlates of fitness in the natural target cells. Finally, on a practical note, our results support close characterisation of virus stocks for experimentation in vitro and in vivo and suggest ways to mitigate unwanted cell culture artefacts, critical for understanding host–pathogen interactions and for the identification of antiviral interventions.

## Figures and Tables

**Figure 1 viruses-14-00325-f001:**
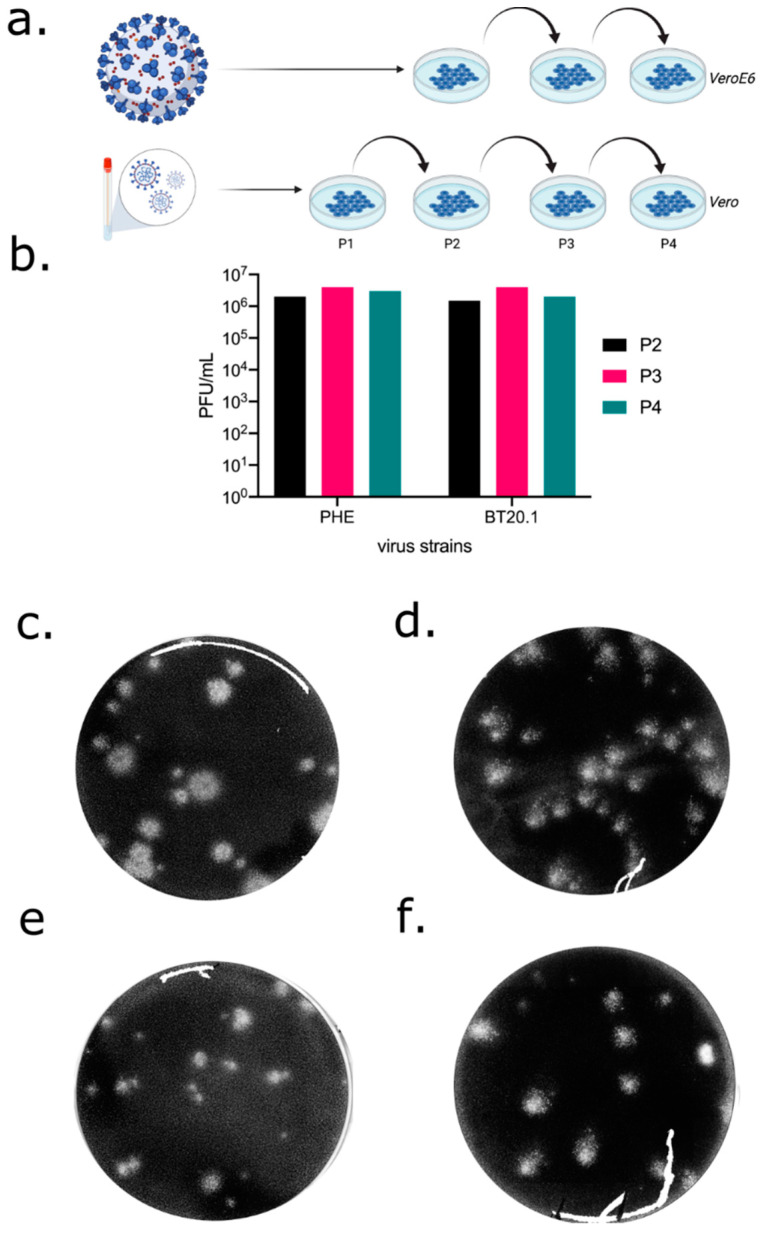
Isolation and passage of PHE and BT20.1 in Vero-derived cells. Schematic of SARS-CoV-2 isolation/serial passage series on VeroE6 or Vero cells for PHE and BT20.1 from isolation to P4 (**a**). Extracellular infectivity titres for stocks generated from P2–P4 VeroE6/Vero passage for PHE and BT20.1 using plaque assay protocol on Vero cells (**b**). Plaque visualisation of PHE (**c**) and BT20.1 (**d**) P4 and P2 (**e**,**f**) on Vero cells. Figures were generated with the aid of BioRender (https://biorender.com/ (accessed on 7 December 2021)).

**Figure 2 viruses-14-00325-f002:**
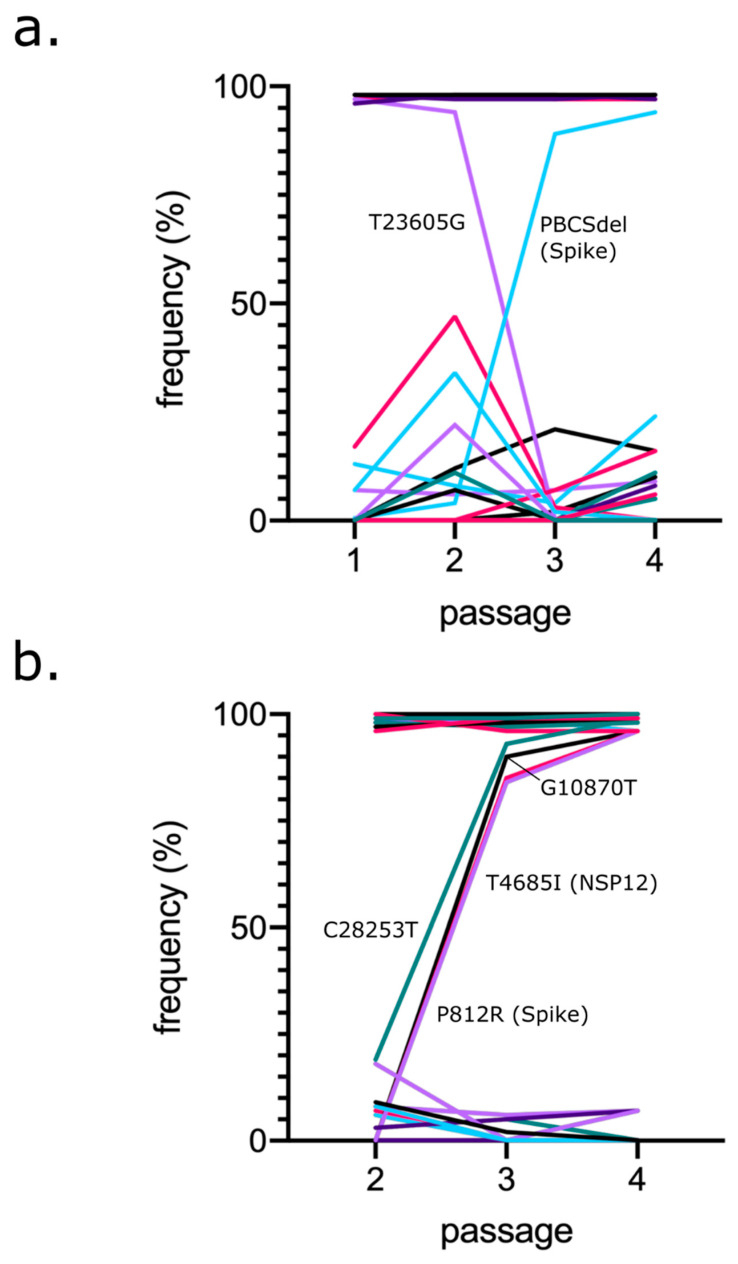
Analysis of PHE and BT20.1 whole-genome sequences during Vero cell passage. Frequency of mutations detected for PHE (**a**) and BT20.1 (**b**) passage series on VeroE6 or Vero cells, relative to the reference sequence (Wuhan-Hu-1). Only sequences from P1-P4 (PHE) and P2-P4 (BT20.1) were analysed to facilitate adequate comparisons. Core changes are found consistently at a high frequency and minor variants are found at consistently a low frequency (e.g., <50%). Only variants that significantly changed in frequency are marked on the graph. Colours do not reflect relationships between variants.

**Figure 3 viruses-14-00325-f003:**
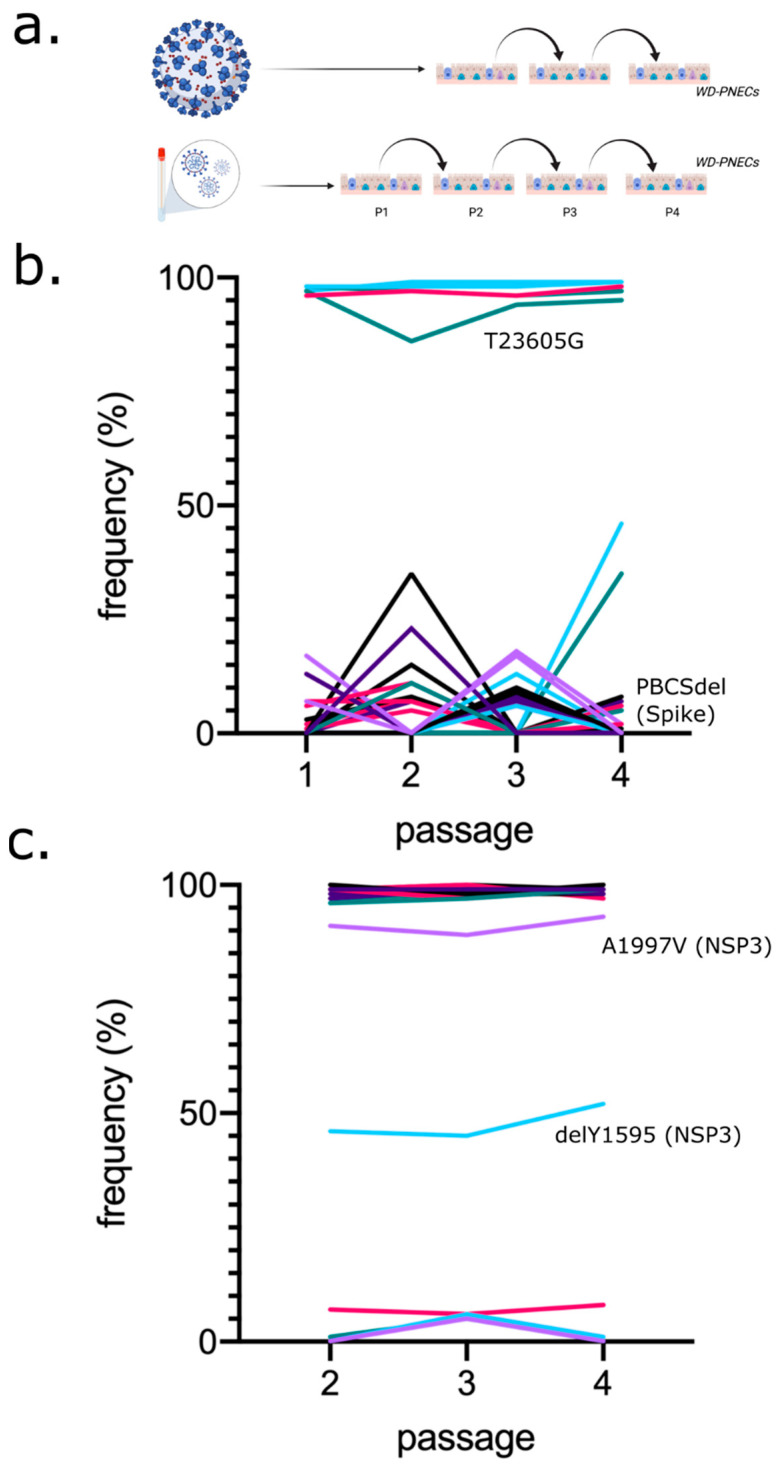
Analysis of PHE and BT20.1 whole-genome sequences during WD-PNECs passage. Schematic of SARS-CoV-2 isolation/passage series on WD-PNECs for PHE and BT20.1 (**a**). Frequency of mutations detected for PHE (**b**) and BT20.1 (**c**) passage series on WD-PNECs, respectively, relative to the reference sequence (Wuhan-Hu-1). Only sequences from P1-P4 (PHE) and P2-P4 (BT20.1) were analysed. PHE P1 is the original stock material obtained and hence is the same sequence as PHE P1 in Figure 2. Core changes were found consistently at a high frequency and minor variants were found at a consistently low frequency (e.g., <50%). Only variants that significantly changed in frequency are marked on the graph. Colours do not reflect relationships between variants. Figures were generated with the aid of BioRender.

**Figure 4 viruses-14-00325-f004:**
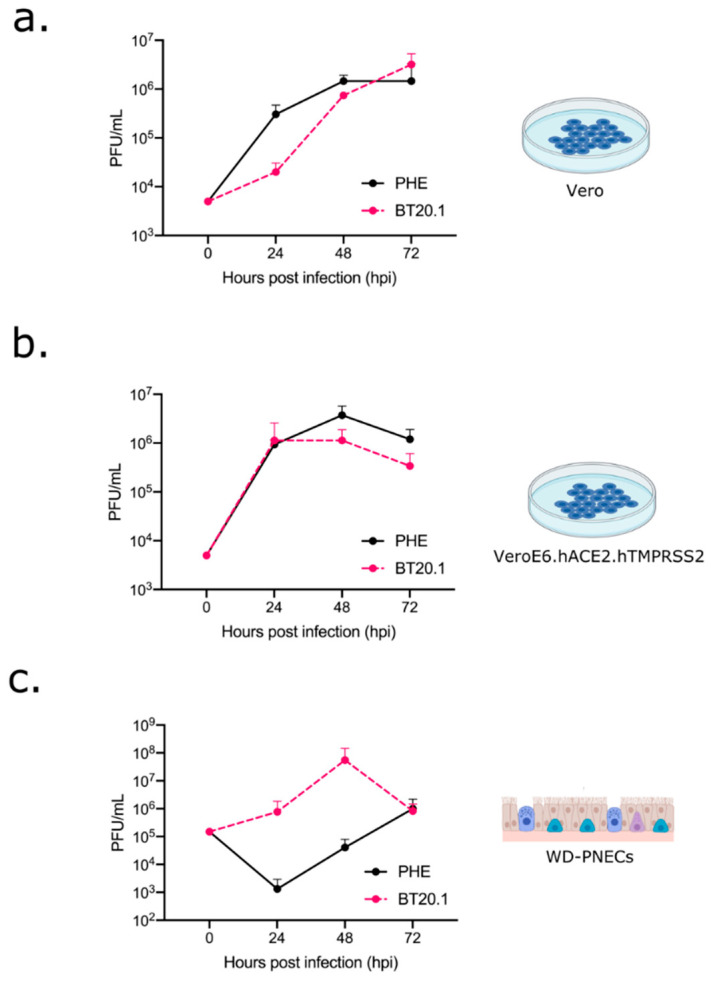
Comparison of PHE P4 (Vero) and BT20.1 P4 (Vero) growth on different cell substrates. Multicycle growth curves (MOI 0.01 for Vero or 0.1 for WD-PNECs) for PHE P4 (VeroE6) and BT20.1 P4 (Vero) on Vero (**a**), VeroE6 cells expressing human ACE2 and human TMPRSS2 (**b**) and adult WD-PNECs from 3 donors (**c**). Titres for Vero-derived cells are shown as means +/− SEM for triplicate wells and are representative of two independent experiments. Titres for WD-PNECs are shown as means +/− SEM for single wells from 3 donors. Data using BT20.1 are presented here as averages from 3 donors but have also been incorporated into a sister paper using separated, individual donor data [26]. Figures were generated with the aid of BioRender.

## Data Availability

Raw sequencing data are available via online repositories (European Nucleotide Archive) linked: 28 July 2021|PRJEB46668 (ERP130880)|“Comparison of Sars2 evolution in vero derived versus primary human airway cells”.

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
