# Peer review of "Comparison of SARS-CoV-2 Evolution in Paediatric Primary Airway Epithelial Cell Cultures Compared with Vero-Derived Cell Lines"

_viruses, 2022, doi:10.3390/v14020325_

Round 1

Reviewer 1 Report

This paper shows very intriguing results that SARS-CoV2 have diversity in evolution in different cell lines in vitro. The difference may come from viral protein and host factor collaboration. In general, the results are clean and conclusion is convincing. while there are some question might need to be answered and clarified.

  1. Is there any other difference in some key host factors between Vero and WD-PNEC besides the spike receptor(ACE2 and TMPRSS2), such as restriction factor involved in innate immunity. Will those restriction factor provide different pressure on viral evolution.
  2. In figure 4C, the viral titer shows significant difference during 24-48hr between PHE and BT20.1 in WD-PNEC cell lines, but become similar after 72hrs. Could the authors provide some information about what might be the cause ? As in Vero and modified Vero cell lines, these two strains have very similar infection trend.
  3. In vivo infection and evolution will have to overcome the adaptive immune response such as antibodies, which is absent in in vitro system, it will be very interesting(but may not be necessary) to see  with some pressure from antibodies, any more mutation could happen or will there be any more diversity in viral rapid evolution in different cell lines 

Author Response

This paper shows very intriguing results that SARS-CoV2 have diversity in evolution in different cell lines in vitro. The difference may come from viral protein and host factor collaboration. In general, the results are clean and conclusion is convincing. while there are some question might need to be answered and clarified.

  1. Is there any other difference in some key host factors between Vero and WD-PNEC besides the spike receptor(ACE2 and TMPRSS2), such as restriction factor involved in innate immunity. Will those restriction factor provide different pressure on viral evolution.

We thank the reviewer for this comment and allowing us to elaborate further. The authors agree with the idea proposed. We do not think that TMPRSS2 necessarily explains everything but it is an important factor, this is because of the known interaction with host restriction factors expressed highly in epithelial cells, such as IFITMs, the viral polybasic cleavage site, and host proteases (e.g. Winstone et al., 2021 https://journals.asm.org/doi/10.1128/JVI.02422-20 .) We added a sentence (line 408) in the discussion with a reference highlighting this.

  1. In figure 4C, the viral titer shows significant difference during 24-48hr between PHE and BT20.1 in WD-PNEC cell lines, but become similar after 72hrs. Could the authors provide some information about what might be the cause ? As in Vero and modified Vero cell lines, these two strains have very similar infection trend.

We thank the reviewer for this comment and would like to firstly note the growth differences (100-1000X at early times) in the primary cultures between strains with PHE P4 (which lacks the PBCS) exhibited a delayed infection compared to BT20.1 P4 (which retains the PBCS), consistent with the maintenance of the PBCS during passage of PHE in primary cells but not during VeroE6 passage. In unmodified Vero cells, BT20.1 with the PBCS has a growth disadvantage compared to PHE lacking the PBCS, which is rescued in VeroE6 cells expressing human ACE2 and TMPRSS2. Furthermore, the growth attenuation of viruses lacking the PBCS is consistent with other studies as well (e.g. Lamers et al., 2021). However, we duly note that PHE P4 and BT20.1 P4 had similar titres at 3dpi, with PHE “catching up” with BT20.1, which had already begun to decline. We thus hypothesise that the PBCS is not essential for replication in the primary cultures but could be more critical during the early, initial stages of infection. Other factors in the following days modify this phenotype, such as differential induction of antiviral factors, success of cell-to-cell spread, and/or high local MOI infection in overcoming antiviral defences. Additionally, as PHE P4 was derived from a mixed population it is possible that over the course of the experiment, viruses retaining the PBCS could be selected for. Our combined data from passage on primary cells, and challenge of cultures with Vero-passaged virus, is consistent with a fitness advantage in primary cells conferred by the PBCS. We have added a sentence regarding this to the results (line 269).

  1. In vivo infection and evolution will have to overcome the adaptive immune response such as antibodies, which is absent in in vitro system, it will be very interesting(but may not be necessary) to see  with some pressure from antibodies, any more mutation could happen or will there be any more diversity in viral rapid evolution in different cell lines  

We agree with the reviewer that addressing the role of antibodies on virus evolution would be a very important study given the role of antibodies as treatments or in vaccines as prophylactics. However, this was beyond the scope of our study, which focused on interactions between paediatric primary cells, and the virus, although other similar studies have addressed this with interesting results (Andreano et al., 2021 https://www.pnas.org/content/118/36/e2103154118 ). Finally, we note some implications of our observations and antibody escape in that we identify a deletion in the NTD, which may enhance virus fitness in vitro (lacking antibodies), and is mapped to neutralising epitopes. Our work thus suggests that antibody-mediated selection is not necessary to drive emergence of escape mutants.

Reviewer 2 Report

The SARS-CoV2 coronavirus has proven to be exceptionally pliable with the identification of various mutations compared to the standard Wuhan strain.  There are three VOCs of note:  alpha, delta and now omicron.  This paper is an attempt to understand what variable of how the interaction of the virus in vitro in host cells might drive the mutation of this particular coronavirus.  The authors approach the question of variability of the SARS-CoV2 virus by determining the sequences of two different viruses in two to three different host cells over low passage in vitro.

The experimental design that is used by the authors is a logical approach to try to define how the virus changes overtime as it replicates in a host.  the available system is to use cell cultures.  The authors are commended to include the primary nasal epithelial cell line in the study as a comparison to the Vero cell line that is permissive for this virus.

Overall the results indicate that SARS-CoV2 replicates in both cell lines but evolves differently.  Evolution of the virus is more frequent in the non-host cell line Vero than the host nasal epithelial cell line.  It is an interesting question as to why the evolution in the adult nasal epithelial cells is restrained?  So the results support the conclusion that host cell factors shape the plasticity of the virus.  A question for the authors to consider is whether the evolution in Vero cells may not be related to pathogenicity but rather a path to attenuation through serial passage?  Sure the passages of virus examined in this study are low but generally as viruses attenuate through cell culture passage, there tends to multiple mutations on various genes.  This may not be the case in a natural host as the virus may be more on a path to increased virulence or transmissibility, thus genetic changes may be more targeted and restrained?  

Overall, this is a good paper that established some possible mutations in the virus in targeted areas that may be important for future VOCs.  It would be interesting if the authors could expand the study by comparing there results to known sequences of the Omicron variant, but this may not be possible.

Some specific comments:

1.Authors it is difficult at time to know if you are discussing Vero or Vero6 cells in the study.  Could you examine the text and be sure this is clear.

2. Difficult to correlate sTable 1 with the information in the results.  Not sure where the variants you discuss are on the Table and how they relate to PHE an BT20.1.  Also perhaps this should not be a sTable but a regular table in the report?

3.Not sure supplementary Figure 1 adds much to the study.  There are differences in plaque sizes and number but this perhaps could just be explained in the narrative.  Same with supplementary Figure 3.  How do these relate to Fig 1c,d, e f?

Author Response

The SARS-CoV2 coronavirus has proven to be exceptionally pliable with the identification of various mutations compared to the standard Wuhan strain.  There are three VOCs of note:  alpha, delta and now omicron.  This paper is an attempt to understand what variable of how the interaction of the virus in vitro in host cells might drive the mutation of this particular coronavirus.  The authors approach the question of variability of the SARS-CoV2 virus by determining the sequences of two different viruses in two to three different host cells over low passage in vitro.

The experimental design that is used by the authors is a logical approach to try to define how the virus changes overtime as it replicates in a host.  the available system is to use cell cultures.  The authors are commended to include the primary nasal epithelial cell line in the study as a comparison to the Vero cell line that is permissive for this virus.

Overall the results indicate that SARS-CoV2 replicates in both cell lines but evolves differently.  Evolution of the virus is more frequent in the non-host cell line Vero than the host nasal epithelial cell line.  It is an interesting question as to why the evolution in the adult nasal epithelial cells is restrained?  So the results support the conclusion that host cell factors shape the plasticity of the virus.  A question for the authors to consider is whether the evolution in Vero cells may not be related to pathogenicity but rather a path to attenuation through serial passage?  Sure the passages of virus examined in this study are low but generally as viruses attenuate through cell culture passage, there tends to multiple mutations on various genes.  This may not be the case in a natural host as the virus may be more on a path to increased virulence or transmissibility, thus genetic changes may be more targeted and restrained?  

We agree with the reviewer and thank them for this question as it is an important point. In fact, in our study, in VeroE6 cells, the virus can readily lose the PBCS, which is required for fitness in primary epithelial cells. Other studies have gone on to map the PBCS as a critical determinant of transmission (Peacock et al., 2021) AND pathogenicity (https://www.nature.com/articles/s41586-021-03237-4 ). These studies were cited in our introduction. Thus, the virus can indeed become attenuated during passage in Vero cells. However, our other strain, BT20.1, while passaged in similar cells, did not lose its PBCS and appeared to not be attenuated in primary cultures. Additionally, one thing we note that in our experiments, we noted that Spike in general was a hotspot for virus evolution.  

Overall, this is a good paper that established some possible mutations in the virus in targeted areas that may be important for future VOCs.  It would be interesting if the authors could expand the study by comparing there results to known sequences of the Omicron variant, but this may not be possible.

We agree with this point and note that our study focussed on pre-VOC variants (D614 and D614G strains). While in general there is no overlap between our major mutations in our evolved populations and VOCs (which is interesting), we note overlap between NTD deletion (that we hypothesise has a replication advantage compared to wild-type) and several variants of concern, including alpha (which we note in the text) and omicron that harbour deletion of 69/70 in Spike. We suggest that this region in NTD of Spike modulates virus replication and could be contributing to the fitness of VOCs, as has been suggested (Meng et al., 20201 https://www.cell.com/cell-reports/pdfExtended/S2211-1247(21)00663-X ). We have a paper in review looking at this in greater detail and we look forward to sharing these results with the community.  Additional overlap between our variants and VOCs is D215G (present in alpha and beta).

Some specific comments:

1.Authors it is difficult at time to know if you are discussing Vero or Vero6 cells in the study.  Could you examine the text and be sure this is clear.

We have updated the text accordingly where it is unclear, as we do refer to both sets of cells throughout the work. In general, we have considered Vero and VeroE6 cells to be similar (referred to as “unmodified Vero cells” or Vero-derived), while distinct from our modified VeroE6 cells expressing human ACE2 and TMPRSS2. We have added a sentence in the methods stating that standard VeroE6 cells were only used for the passage of PHE.

  1. Difficult to correlate sTable 1 with the information in the results.  Not sure where the variants you discuss are on the Table and how they relate to PHE an BT20.1.  Also perhaps this should not be a sTable but a regular table in the report?

We thank the reviewer for this point given the importance of the data in the table. We have updated the table and its legend to make it clearer and added: ““Variant” mutations, which are shown in the main figures, have been highlighted in bold text.” Given its size and complexity (as well as overlap with graphs in figures for the key data) we have kept it as supplementary to help interpretation of critical results. Additionally, all raw data are available via online repositories (European Nucleotide Archive) linked: 28-Jul-2021 | PRJEB46668 (ERP130880) | Comparison of sars2 evolution in vero derived versus primary human airway cells

3.Not sure supplementary Figure 1 adds much to the study.  There are differences in plaque sizes and number but this perhaps could just be explained in the narrative.  Same with supplementary Figure 3.  How do these relate to Fig 1c,d, e f?

sFig1 and 3 highlights differences in plaque morphology/fusogenicity between strains and aids description of biological differences related to spike activity between strains. As journals typically do not like “data not shown” we have opted to retain this observation but keep it only as supplement alongside mention in the text narrative. sFig1 is a magnified image of plaques from Fig 1.  

Reviewer 3 Report

This is a very well written article, I don't have any major comments.

Author Response

We are delighted that the third reviewer did not find any issues to comment on.